# Monodisperse SiO_2_ Spheres: Efficient Synthesis and Applications in Chemical Mechanical Polishing

**DOI:** 10.3390/nano15090665

**Published:** 2025-04-27

**Authors:** Jinlong Ge, Yu Cao, Hui Han, Xiaoqi Jin, Jing Liu, Yuhong Jiao, Qiuqin Wang, Yan Gao

**Affiliations:** 1School of Materials and Chemical Engineering, Bengbu University, 1866 Cao Shan Road, Bengbu 233030, China; jwh@bbc.edu.cn; 2Anhui Provincial Engineering Research Center of Silicon-Based Materials, Bengbu University, 1866 Cao Shan Road, Bengbu 233030, China; l3167526736@163.com (X.J.); wangqiuqin000@163.com (Q.W.);; 3Bengbu Zhongheng New Material Technology Co., Ltd., 751 Donghai Road, Bengbu 233030, China; caoyu8086@163.com (Y.C.); 13956386350@163.com (H.H.); liujing2017@163.com (J.L.)

**Keywords:** Stöber method, SiO_2_ sphere, chemical mechanical polishing, planarization, SiO_2_ slurry

## Abstract

The atomic level polishing of a material surface affects the accuracy of devices and the application of materials. Silica slurries play an important role in chemical mechanical polishing (CMP) by polishing the material surface. In this study, an efficient and controllable Stöber approach was developed to synthesize uniform monodisperse silica spheres with different cationic surfactants. The obtained silica spheres exhibited a regular shape with a particle size of 50–150 nm and were distributed evenly and narrowly. The highest surface specific area of the silica spheres was approximately 1155.9 m^2^/g, which was conducive to the polish process. The monodisperse SiO_2_ spheres were applied as abrasives in chemical mechanical polishing. The surface micrographs of silicon wafers during the CMP process were studied using atomic force microscopy (AFM). The results demonstrated that the surface roughness Ra values reduced from 1.07 nm to 0.979 nm and from 1.05 nm to 0.933 nm when using a CTAB-SiO_2_ microsphere as an abrasive. These results demonstrate the advantages of monodisperse SiO_2_ spheres as abrasive materials in chemical mechanical planarization processes.

## 1. Introduction

Chemical mechanical polishing (CMP) is an essential technology for the atomic-level planarization of a diverse variety of materials, such as silicon wafers, silicon carbide, sapphire, copper, gallium nitride, and glass [1,2,3]. In a typical CMP process, there are many influencing factors involved in the polishing planarization process, including pad properties, slurry characteristics, and processing conditions. It is well known that chemical and mechanical mechanisms are two important mechanisms in the polishing process [4,5,6]. Investigations of the variables in the CMP process have found that solid loading, particle size and distribution, modulus, hardness, asperity sizes and distribution, down pressure, and velocity are responsible for material removal [7,8,9].

Currently, mechanically active silica (SiO_2_), ceria (CeO_2_), alumina (Al_2_O_3_) [10], and zirconia (ZrO_2_) are widely used in the preparation of abrasives [11]. SiO_2_ spheres have been applied in many fields, such as nanomedicine, coating, excipients, and additives [12,13,14]. In addition, the narrow diameter distribution of monodisperse SiO_2_ spheres can reduce the agglomeration of abrasives so that the SiO_2_ sphere can be used as an abrasive to reduce the surface roughness of fine instruments [15]. Soft SiO_2_ abrasives cause small scratches on wafers and ensure machining quality. Monodisperse silica spheres represent a novel, environmentally friendly slurry, suitable for the CMP of high-performance device wafers. This innovation aims to mitigate environmental pollution and minimize potential health hazards for operators [16,17,18,19]. The surface topography can be detected via SEM and AFM characterization techniques. The physical properties and surface topography of spheres can be obtained by SEM [20]. For AFM, the surface morphologies are provided by the interaction force between the probe tip and surface [21], as well as the synthesis, development, and characterization of nanoparticles in the field of surface and interface.

Amir et al. successfully developed a malic acid-functionalized superparamagnetic iron oxide nanoparticle-based nano-abrasive with an 8–26 nm narrow particle size. The narrow diameter distribution and low degree of aggregation provided a significantly lower surface roughness and a high material removal rate [22,23,24]. Silica slurries show good planarity, high polishing rate, and high selectivity, which have the greatest influence and make sense in the CMP optimization process. The regular ball-like shapes of SiO_2_ abrasives cause less scratch damage and defects during CMP [25,26,27]. Substantial research efforts have been conducted to improve the MRR of silicon wafers with minimally damaged surfaces during the CMP process based on silica abrasives [28].

Chen et al. estimated the interaction forces between abrasive nanoparticles and substrate surfaces. The chemical tooth was processed with ceria particles, which formed a Si-O-Ce bond between the ceria particles and the sample surface [29]. Chen et al. prepared parallel-channel hexagonal mesoporous silica (H-mSiO_2_) particles with CeO_2_ nanoparticles attached [30]. The H-mSiO_2_-CeO_2_ particles used as abrasives showed a reduced surface roughness, a low topographical variation, and an improved removal rate.

Chen et al. used the good uniformity and dispersity of 30–140 nm SiO_2_ nanospheres as functionalized abrasives [31]. The abrasives played a key role and achieved surfaces with almost no damage and atomic-level roughness, thus avoiding surface scratches commonly caused by particle agglomerations in slurries [32].

Shi et al. compared colloidal silica of different sizes, namely 5 nm to 20 nm as small nano-abrasives, 20 nm to 100 nm as normal abrasives, and 60 nm to 130 nm as a large abrasive slurry. The results showed that the normal- and large-sized abrasives could produce atomic step-terrace structures, with both having high CMP efficiency and perfect planarization quality [30]. It was confirmed that differences in terrace structures were caused by the local miscut variation of the surface of the wafer and not by the CMP technique [33].

In this work, we successfully synthesized monodisperse SiO_2_ nanospheres using the Stöber method, obtaining a controllable 50–150 nm particle size and distribution. Furthermore, the CMP performance of the silica colloids with controllable sizes was studied. In addition, the CMP technical method for producing SiO_2_ containing polishing slurry with abrasive wafers of different sizes was proposed. The obtained SiO_2_ spheres used as abrasives are conducive to surface modification and fine processing.

## 2. Materials and Methods

### 2.1. Materials

Tetraethylorthosilicate (TEOS, A.R.), an ammonia aqueous solution (28 wt%), ethanol (A.R.), cetyltrimethylammonium chloride (CTAC, A.R.), cetyltrimethylammonium bromide (CTAB, A.R.), octadecytrimethyl ammonium bromide (STAB, A.R.), cetyldimethyl benzyl ammonium chloride (HDBAC, A.R.), triethanolamine (TEA, A.R.), NH_4_F, and NaOH were purchased from Sinopharm Chemical Reagent Co., Ltd. (Shanghai, China) The obtained chemicals were used as received without further purification. Deionized water was used in all experiments.

### 2.2. Preparation of Silica Spheres

The Stoeber method was employed for the synthesis of silica gel [34]. Typically, a 50 mL round-bottom flask equipped with a stirring rod rotating at 750 rpm was utilized. Solution A consisted of 100 mg NH_4_F (2.7 mmol), 21.7 g H_2_O (1.12 mol), and 2.41 mL of 25% aqueous CTAC (1.83 mmol) in the flask, followed by heating at 60 °C. In a sealed polypropylene test tube, 14.35 g of TEA (97 mmol) and 2.06 mL of TEOS (9.3 mmol) were heated at 90 °C for 30 min without mixing with each other initially. Subsequently, solution B was rapidly added to solution A and stirred, after which the oil bath was removed and the reaction solution was allowed to gradually cool to room temperature while continuously stirring overnight. The following day, after a duration of 12 h, the solution received an addition of 50 mL ethanol before being transferred into two separate centrifuge tubes with a volume of fifty milliliters each; these tubes were then centrifuged at room temperature for twenty minutes at a speed of twenty thousand revolutions per minute (rpm). The supernatant liquid was decanted, and the sample was re-suspended in thirty milliliters of ethanol using both spoon agitation and sonication for ten minutes before undergoing another round of centrifugation [35]. CTAB-SiO_2_, STAB-SiO_2_, and HDBAC-SiO_2_ were synthesized by substituting equal amounts of CTAC with CTAB, STAB, or HDBAC, respectively.

### 2.3. Polishing Tests and Evaluations

CMP experiments on silicon wafers were performed using a UNIPOL-300 CMP machine with a Rodel porous polyurethane pad (Shenyang Kejing Instrument Co., Ltd., Shenyang, China). Force–volume images were scanned within a 3 × 3 µm^2^ area with a resolution of 10 × 10 pixels. The 1 wt% solid content SiO_2_ abrasive particles were dispersed into deionized water and sonicated for 30 min before use. The slurry pH value was altered to 8.5–8.6 using 0.1 M of the NaOH solution. The feed rate of the polishing liquid was 180 mL/min, the pressure was 6 psi, and the table–platen speed was 70 rpm. After polishing for 2 h, the substrates were cleaned with ultrasonic treatment in deionized water. Finally, they were dried with a stream of nitrogen prior to surface analyses. A schematic representation of the complete synthesis route for the monodisperse SiO_2_ spheres and the polishing process of the wafers with SiO_2_ sphere abrasives is displayed in Figure 1.

### 2.4. Characterization

Fourier transform infrared spectra (FT-IR) were measured using KBr pellets on a Nicolet iS10 analyzer (Thermo Fisher Scientific, Waltham, MA, USA) in the range of 4000–400 cm^−1^. The samples were treated using the potassium bromide pellet technique before testing. X-ray diffraction analyzed the crystal structures of the samples based on a Rigaku Smart Lab SE with Cu Ka radiation (λ = 1.54056 A). The diffraction data were collected over an angle range of 5–80° with a step size of 0.02 at 35 kV and 20 mA. A thermogravimetric analysis was performed in a temperature range of 30–900 °C under nitrogen at a heating rate of 10 °C min^−1^ with a Netzsch STA 2500 Regulus analyzer (NETZSCH-Gerätebau GmbH, Selb, Germany). Nitrogen sorption–desorption isotherms were carried out at 77 K using a Micromeritics ASAP 3020 sorption meter (Micromeritics Instrument Corporation, Norcross, GA, USA). The Brunauer–Emment–Teller method was applied to analyze the surface area based on the N_2_ isotherm data. An X-ray photoelectron spectrometer (Kratos Analytical Axis Ultra, Kratos Analytical, Manchester, UK) equipped with a monochromatic Al Kα source was utilized to evaluate the elemental composition. The structure and morphology of the samples were investigated with a Zeiss Sigma 300 scanning electron microscope (Carl Zeiss AG, Oberkochen, Germany) at an acceleration voltage of 15 kV, and the probe current was 50 pA. The morphology and surface roughness of the polished silicon wafers were characterized using atomic force microscopy with Multimode 8, Bruker, Santa Barbara, CA, USA. The samples were measured using the contact mode in the air. The tip radius was 10 nm, and the tip height was 17.5 μm.

## 3. Results

### 3.1. Structural and Textural Features

The XRD patterns of the SiO_2_ samples were measured and are shown in Figure 2. As can be seen, the strong and broad diffraction peak at 2θ of 23° is in good agreement with the position of pure amorphous silica [36]. This result means that no discernible long-range order in the pore arrangement exists in the SiO_2_. These peaks indicate that the silica microspheres are successfully synthesized via the sol–gel method.

The composition of the silica spheres was established via FT-IR spectra, as shown in Figure 3. The strong band at 3442.2 cm^−1^ can be assigned to the absorption of the H_2_O of the silica spheres. The absorption band appearing at 2870.4 cm^−1^ indicates the stretching vibration of -CH_3_ in CTAB. The peak at 1630.5 cm^−1^ belongs to the bending vibration of the -OH groups. The peak with wavenumbers of 1404.9 and 1383.8 cm^−1^ belongs to -CH_3_ and -CH_2_ symmetric bending vibrations, respectively. The peak at 970.9 cm^−1^ is associated with the bending vibration of Si-OH. The obvious bands located at 1054.7 and 457 cm^−1^ can be assigned to the stretch vibration bands of the Si-O-Si bond [37]. This indicates that SiO_2_ is hydrophilic and that there exist -OH groups on its surface.

The TGA curves of SiO_2_ obtained under different preparation conditions are shown in Figure 4. The first slight weight loss of 8% below 120 °C is ascribed to the dissociation of absorbed water. The weight loss of 15% at 120 °C to 550 °C can be attributed to the degradation of organic parts [38].

During the synthesis of the silica spheres, the Brunauer–Emmett–Teller specific surface area was monitored using nitrogen adsorption–desorption isotherms, which are shown in Figure 5. The silica spheres show typical type IV adsorption isotherms, which are normally attributed to the characteristics of ordered mesoporous channels. There exists a pore condensation process at the relative pressure range of P/P_0_ = 0.3–0.4. Specifically, the BET surface areas of CTAC-SiO_2_, CTAB-SiO_2_, STAB-SiO_2_, and HDBAC-SiO_2_ comparably decreased from 1155.9, 1059.0, and 1119.2 to 796.9 m2 g^−1^ (Table 1). The pore sizes of these samples reduced from 2.4, 2.4, and 2.38 to 2.35 nm, as shown in Figure 6. The pore size did not change significantly with the different surfactants.

The morphology and particle size of the SiO_2_ microspheres were confirmed using SEM, as shown in Figure 7. All samples had a regular spherical shape [39]. The surfaces of CTAB-SiO_2_ and CTAC-SiO_2_ were smoother than those of the other surfactants with SiO_2_. The particle sizes of CTAC-SiO_2_, CTAB-SiO_2_, STAB-SiO_2,_ and HDBAC-SiO_2_ were summarized by a nano measurer according to the SEM images and are presented in Figure 7. The average particle sizes of the SiO_2_ microsphere samples were about 50–150 nm.

The chemical changes and valence states were evaluated by XPS. As shown in Figure 8, the elements Si2p, O1s, and C1s were detected in the samples. The spectrum of the SiO_2_ microspheres shows strong peaks at 531.9 eV, corresponding to the binding energy of O1s [40,41]. The spectrum shows a BE of 533.19 eV and 532.94 eV, corresponding to Si-O. The electronic binding energy of the C1s peak at 284.6 eV corresponds to the C–C bond [42]. The peaks seated at 283.4 and 286.3 eV correspond to C-Si and H-C bonding. The peaks of 103.75 and 103.90 eV were assigned to Si2p. The small shoulder peak of 102.9 eV corresponds to H_6_C_2_Si_2_O_3_. The slurry actively reacts with the oxide surface, leading to the dissolution of Si–O bonds into H–C–O–Si bonds [43]. This hydro-carbonated surface of the oxide film was easier to remove during the mechanical part of the CMP process.

### 3.2. Polishing Performance of SiO_2_ Microspheres

As shown in Figure 9, AFM micrographs show that by using the obtained composite particles as an abrasive, a flat, smooth surface can be obtained without significant scratches or residual particles [44,45]. The surface quality of the silicon wafers after polishing with the CTAB-SiO_2_, CTAC-SiO_2_, STAB-SiO_2_, and HDBAC-SiO_2_ particles can be investigated through their surface topographies. Representative two-dimensional (2D) AFM height images are obtained, and the 3D topography of the wafers is further determined. The wafer micrographs show a fixed test point with a 10 µm × 10 µm area. In AFM height images, the bright points are the high areas, while the dark spots are the low areas [46]. The uniform colors in the AFM height images suggest a flatness and smooth surface [3]. By comparing the AFM 2D images of the wafers polished with the CTAB-SiO_2_, CTAC-SiO_2_, STAB-SiO_2_, and HDBAC-SiO_2_ abrasives, it is found that the most uniform color can be seen in the image of the wafer polished using CTAC-SiO_2_, while the worst is the wafer polished using HDBAC-SiO_2_. Moreover, the sequence of Ra values calculated using their height data was 1.41, 2.14, 2.43, and 2.52 nm, and the Rq values were 1.01, 1.49, 1.65, and 1.73 nm, respectively [47].

Scratches, as well as other microdefects, can hardly be observed [48]. The particle size of silica abrasives plays an essential role during the CMP process [49]. It can be clearly seen that the CTAB-SiO_2_ and CTAC-SiO_2_ abrasives show more regular round shapes and remain independent for each particle [50]. Some mechanical scratches and cracks are distributed deeply on the HDBAC-SiO_2_ abrasives. The particle size of HDBAC-SiO_2_ is about 140 nm, which is larger than that of the other SiO_2_ abrasives [51,52], which is attributed to the higher Ra values, and perhaps the poorer surface quality results from some indentations of embedded SiO_2_ with a large size or stiffness [47].

The surfactant group may be adsorbed on the surface of the SiO_2_ abrasive under the action of electrostatic attraction. The Si-OH in the slurry interacts with the silicon wafer surface in a similar fashion to the Si-OH on the SiO_2_ grain surface, both in the form of bridge bonds, thereby reducing the breakage bond energy of the Si–Si bonds inside the silicon [53,54]. It is noteworthy that optimal polishing performance can be achieved by controlling the balance between the chemical effect and the mechanical effect [55].

AFM images of 10 µm × 10 µm regions were applied to further explore the surface features of the wafers after polishing with the monodisperse SiO_2_ spheres. As shown in Figure 10a–d, the wafer images exhibit a rough surface before the CMP process. The corresponding surface roughness Ra values of the original wafers calculated using their height data are 1.07 and 1.05 nm, and the Rq values are 1.47 and 1.43 nm, respectively. After polishing with the CTAB-SiO_2_ and CTAC-SiO_2_ microspheres, the wafers’ Ra values reduce from 1.07 nm to 0.979 nm and from 1.05 nm to 0.933 nm, and their Rq values reduce from 1.47 nm to 1.31 nm and from 1.43 nm to 1.26 nm, respectively (Figure 10e–h). Moreover, the wafers have no mechanical scratches or cracks after grinding and polishing using the prepared monodisperse SiO_2_ spheres as abrasives [56,57]. This demonstrates that the monodisperse SiO_2_ sphere abrasives can steadily maintain their ultra-precision machining ability. Therefore, it can be considered that the CMP performance achieved by the monodisperse SiO_2_ sphere abrasives is stable, with valuable practical applications.

The material removal mechanism of CMP is complex. Although we demonstrated that the prepared monodisperse SiO_2_ spheres significantly contributed to improved CMP performance, many issues worthy of scientific investigation still exist. The interfacial interaction between the abrasive particles and the substrate surface and the physicochemical properties, adhesion, friction, and wear behavior of the monodisperse SiO_2_ spheres were examined, and the results showed that the monodisperse SiO_2_ spheres can be dramatically improved by optimizing abrasive structures and polishing parameters.

## 4. Conclusions

In summary, SiO_2_ microspheres with a controllable size of 50–150 nm were successfully synthesized using the Stöber method with a series of cationic surfactants. Spherical SiO_2_ exhibited an improved surface quality in the chemical mechanical polishing process. The SiO_2_ microspheres exhibited a high surface area of 1155.9 m^2^/g. The monodisperse SiO_2_ spheres were successfully applied as abrasives in chemical mechanical polishing. Surface micrographs of silicon wafers during the CMP process were studied using AFM. The results demonstrate that the surface roughness Ra values reduced from 1.07 nm to 0.979 nm and from 1.05 nm to 0.933 nm when using the CTAB-SiO_2_ microsphere as an abrasive. The polishing results indicate that the SiO_2_ abrasives can achieve substantial improvement in surface planarization and will provide guidance on precision machining for other key materials. The prepared monodisperse SiO_2_ sphere abrasives promote the development of CMP technology and meet the increasing demand for ultra-precision surfaces for industrial applications.

## Figures and Tables

**Figure 1 nanomaterials-15-00665-f001:**
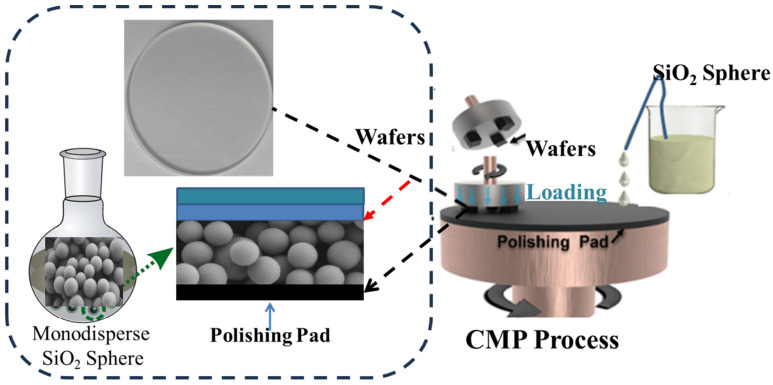
Schematic representation of polishing process of wafers with monodisperse SiO_2_ sphere abrasives.

**Figure 2 nanomaterials-15-00665-f002:**
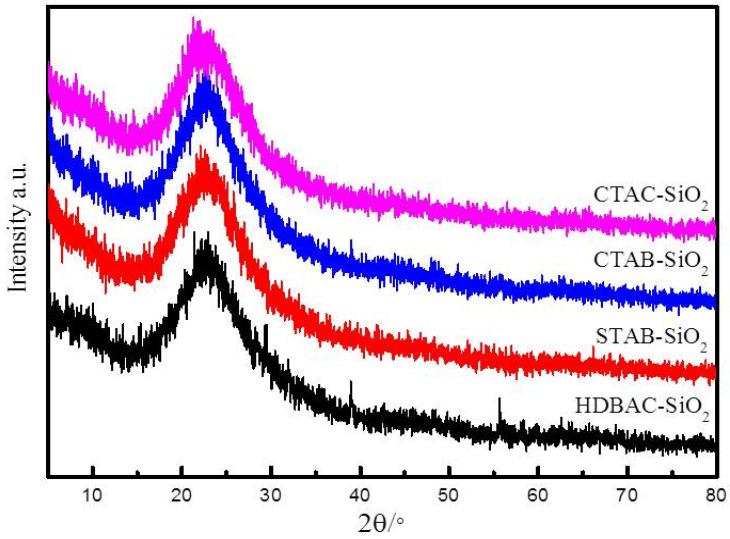
XRD patterns of SiO_2_ microspheres.

**Figure 3 nanomaterials-15-00665-f003:**
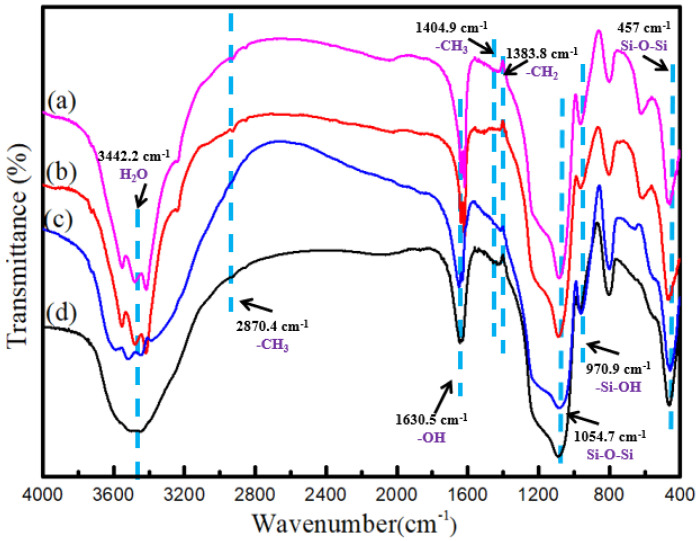
FTIR spectra of SiO_2_ microspheres: (a) CTAC-SiO_2_, (b) CTAB-SiO_2_, (c) STAB-SiO_2_, and (d) HDBAC-SiO_2_.

**Figure 4 nanomaterials-15-00665-f004:**
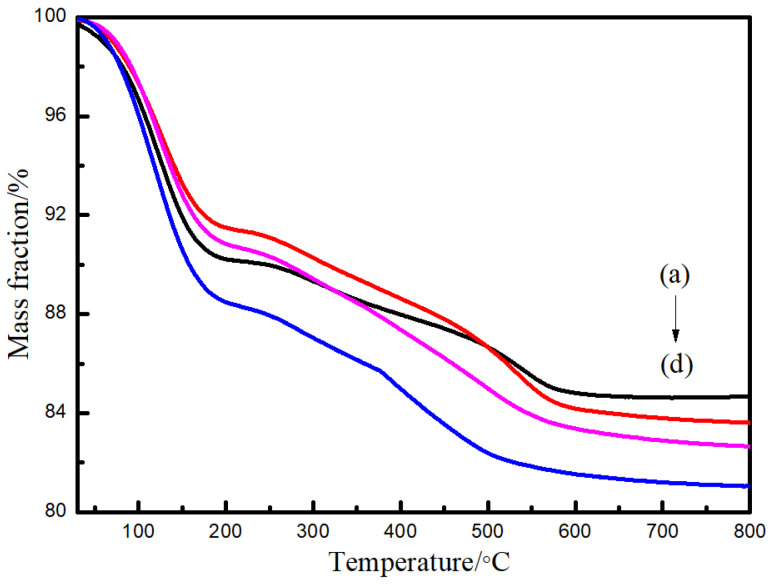
TGA spectra of SiO_2_ microspheres: (a) CTAC-SiO_2_, (b) CTAB-SiO_2_, (c) STAB-SiO_2_, and (d) HDBAC-SiO_2_.

**Figure 5 nanomaterials-15-00665-f005:**
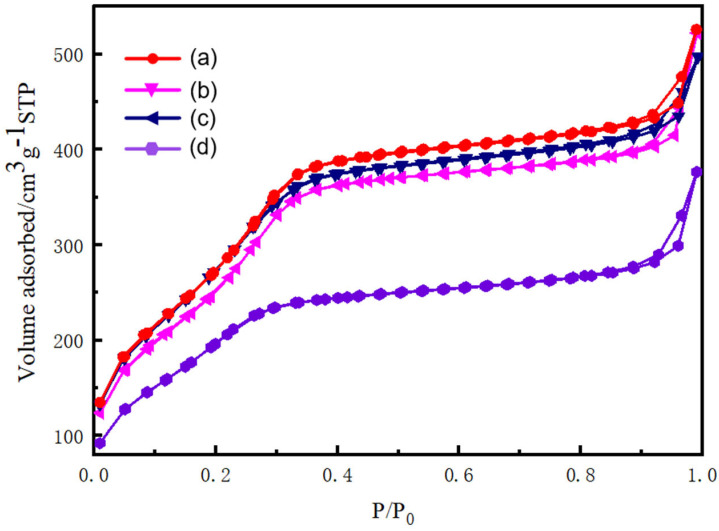
N_2_ adsorption–desorption isotherms of SiO_2_ microspheres: (a) CTAC-SiO_2_, (b) CTAB-SiO_2_, (c) STAB-SiO_2_, and (d) HDBAC-SiO_2_.

**Figure 6 nanomaterials-15-00665-f006:**
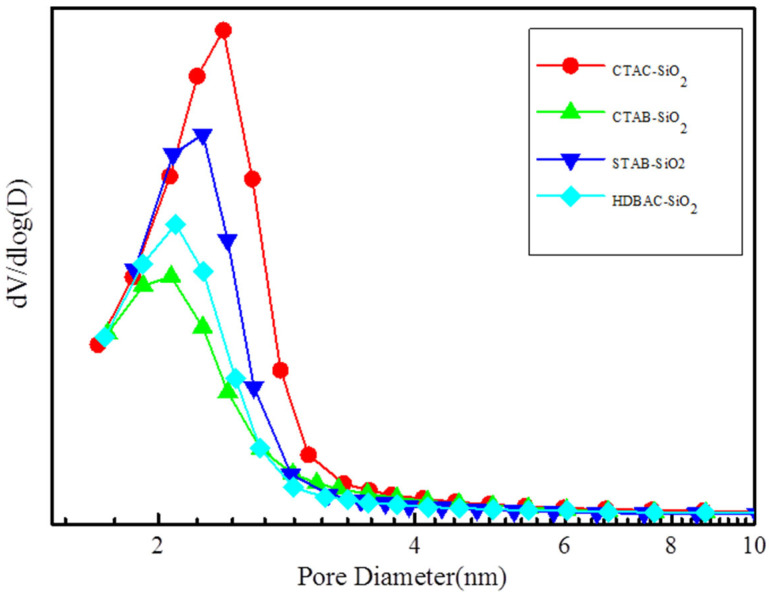
The corresponding pore size distributions.

**Figure 7 nanomaterials-15-00665-f007:**
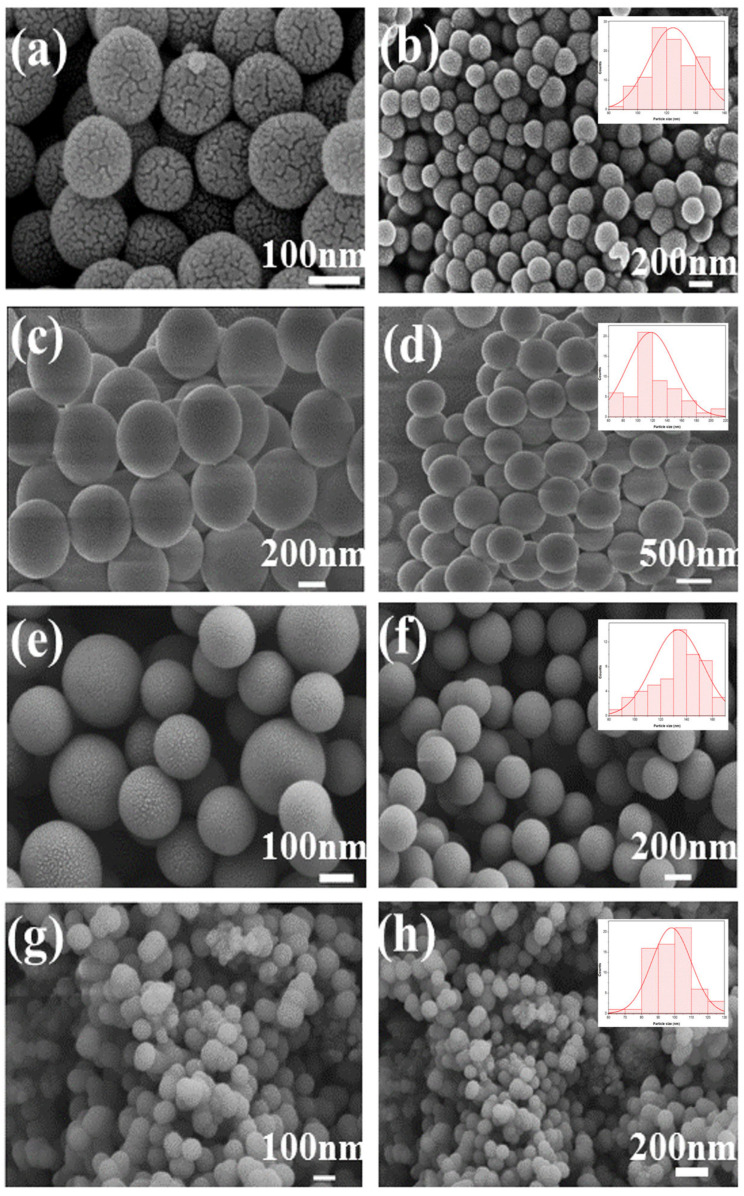
SEM images: (**a**,**b**) CTAC-SiO_2_, (**c**,**d**) CTAB-SiO_2_, (**e**,**f**) STAB-SiO_2_, and (**g**,**h**) HDBAC-SiO_2_.

**Figure 8 nanomaterials-15-00665-f008:**
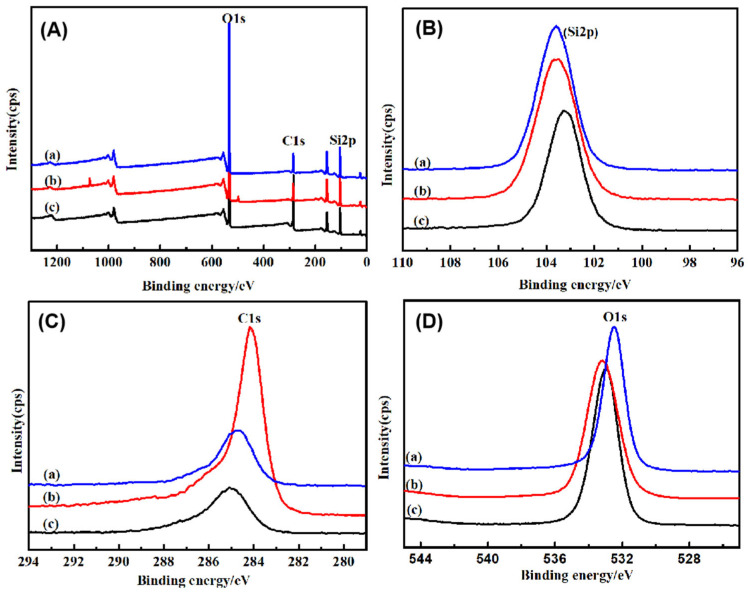
XPS spectra of SiO_2_ microspheres: (a) CTAC-SiO_2_, (b) CTAB-SiO_2_, and (c) STAB-SiO_2_. (**A**) Full XPS survey; (**B**) Si2p; (**C**) C1s; (**D**) O1s.

**Figure 9 nanomaterials-15-00665-f009:**
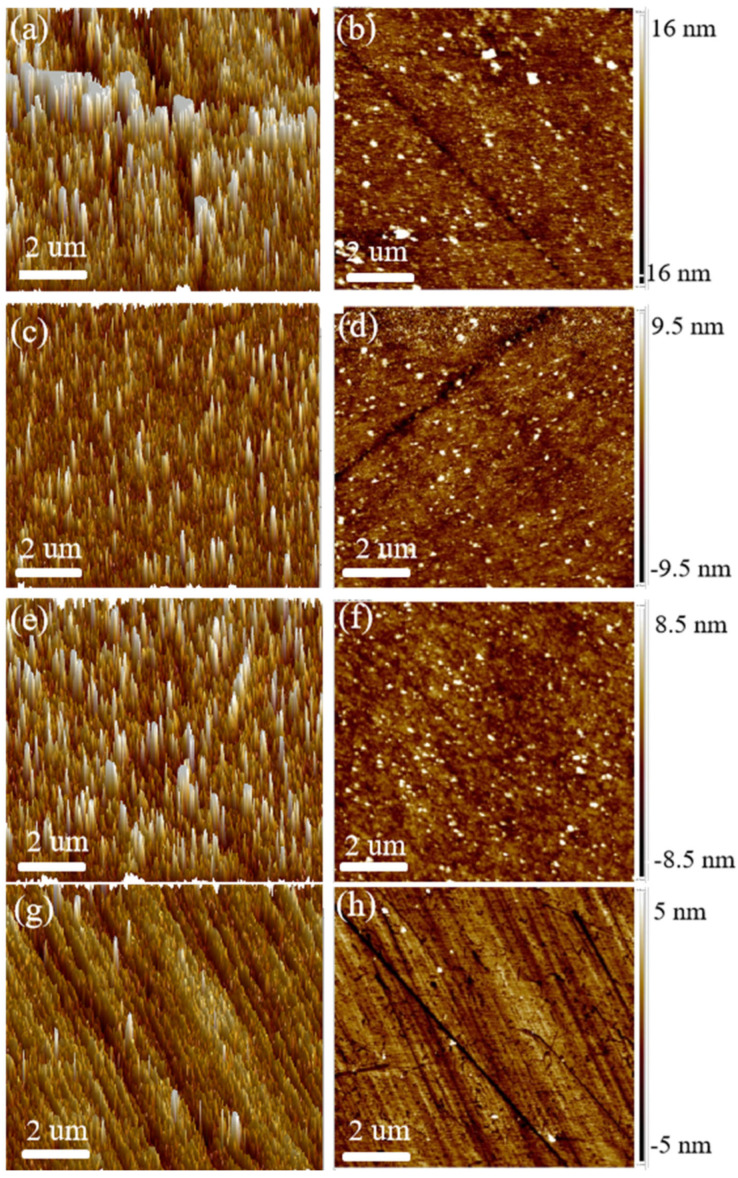
A range of 3D and 2D AFM images of the SiO_2_ microspheres after CMP: (**a**,**b**) CTAC-SiO_2_, (**c**,**d**) CTAB-SiO_2_, (**e**,**f**) STAB-SiO_2_, and (**g**,**h**) HDBAC-SiO_2_.

**Figure 10 nanomaterials-15-00665-f010:**
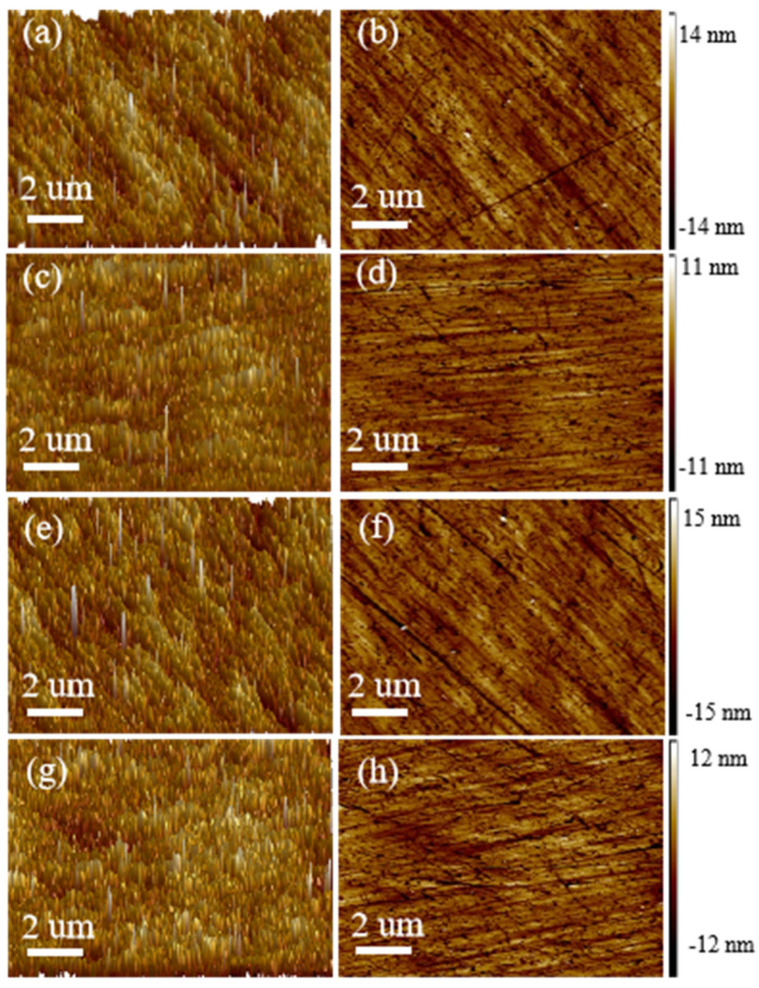
A range of 3D and 2D AFM images of the CTAB-SiO_2_ (**a**,**b**) before and (**c**,**d**) after the wafer surfaces. AFM images of the CTAC-SiO_2_ microsphere (**e**,**f**) before and (**g**,**h**) after the wafer surfaces.

**Table 1 nanomaterials-15-00665-t001:** The structure parameters of all related samples.

Samples	S_BET_ (m^2^ g^−1^)	Pore Size (nm)
CTAC-SiO_2_	1155.9	2.40
CTAB-SiO_2_	1119.2	2.40
STAB-SiO_2_	1059.0	2.38
HDBAC-SiO_2_	796.9	2.35

## Data Availability

Data will be made available on request.

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
