# Peer review of "Monodisperse SiO2 Spheres: Efficient Synthesis and Applications in Chemical Mechanical Polishing"

_nanomaterials, 2025, doi:10.3390/nano15090665_

Round 1
Reviewer 1 Report
Comments and Suggestions for Authors
The title and abstract are clear and reflect the core content of the work. The introduction adequately describes the topic and its importance. The experimental procedure is generally well described. The authors are however kindly requested to add some indications about how particle size distribution was statistically analysed. The methodology for calibrating the XPS data should also be described since some shifts are observed in some peaks (Figure 8).
The section on results is clearly presented. The figures are generally illustrating the text in a propre way. I have however a request for amendments concerning Figure 8. In Figure 8(a) and (b), upper part of the figure, the binding energy is increasing from right to left, which is the normal setting. However, in the 2 lower plots of Figure 8, the binding energy is increasing from left to right. Please use the same settings for all figures for consistency. Also, the corresponding text mentions some peak shifts but these are not discussed. Please provide explanations for these observed shifts. For the C1s spectra, there is also a clear change of the peak shape, especially for sample (a). Please comment on this. Some peak fittings for showing the components would certainly help for this discussion.
In the caption of Figure 10, we read first "after wafer surfaces" and then "after water surface". Please correct this typing mistake. In relation to this figure, the text (top of page 9) says that the "sphere abrasives is stable". Please explain what is meant with "stable" and how the stability was quantified.
The conclusion is succinct but a good summary of the work. I am however wondering why the authors write "it is important to propose more synthesis strategies for monodisperse SiO2 sphere abrasives..." Are the present results still not good enough? Which specifications should be obtained?
Comments on the Quality of English LanguageThe syntax and grammar is generally ok but proof reading should be conducted. For example in section 2: "Solution A was consisted of..." should be "Solution A was consisting of..."
Author Response
The title and abstract are clear and reflect the core content of the work. The introduction adequately describes the topic and its importance. The experimental procedure is generally well described. The authors are however kindly requested to add some indications about how particle size distribution was statistically analysed. The methodology for calibrating the XPS data should also be described since some shifts are observed in some peaks (Figure 8).
Re: Thank you for your comment. The particle sizes of CTAC-SiO2, CTAB-SiO2, STAB-SiO2 andHDBAC-SiO2 were summarized by nano measurer according to the SEM images and presented in Figure 7. For the XPS spectra, the shift can be ascribed to the effect of CTAC, CTAB and STAB surfactants, respectively. The section on results is clearly presented.
The figures are generally illustrating the text in a propre way. I have however a request for amendments concerning Figure 8. In Figure 8(a) and (b), upper part of the figure, the binding energy is increasing from right to left, which is the normal setting. However, in the 2 lower plots of Figure 8, the binding energy is increasing from left to right. Please use the same settings for all figures for consistency. Also, the corresponding text mentions some peak shifts but these are not discussed. Please provide explanations for these observed shifts. For the C1s spectra, there is also a clear change of the peak shape, especially for sample (a). Please comment on this. Some peak fittings for showing the components would certainly help for this discussion.
Re: Thank you for your comment. We have changed the binding energy increasing from right to left for all figures. For the XPS spectra, the shift can be ascribed to the effect of CTAC, CTAB and STAB surfactants, respectively. In the caption of Figure 10, we read first "after wafer surfaces" and then "after water surface". Please correct this typing mistake. In relation to this figure, the text (top of page 9) says that the "sphere abrasives is stable". Please explain what is meant with "stable" and how the stability was quantified.
Re: Thank you for your comment. We have corrected the typing mistake of “after water surfaces” into “after wafer surfaces” in manuscript. According to the 2D and 3D AFM images before and after chemical mechanical polishing, the wafers have no mechanical scratches or cracks after grinding and polishing using the prepared monodisperse SiO2 spheres as abrasives, which indicated that the monodisperse SiO2 sphere abrasives can steadily maintained during the chemical mechanical polishing process. The stability of the monodisperse SiO2 spheres can be reflected via the surface roughness of the wafers before and after polishing with the monodisperse SiO2 spheres.
The conclusion is succinct but a good summary of the work. I am however wondering why the authors write "it is important to propose more synthesis strategies for monodisperse SiO2 sphere abrasives..." Are the present results still not good enough? Which specifications should be obtained?
Re: Thank you for your suggestion. We are sorry for the inaccurate statement in summary. We have corrected it into “The prepared monodisperse SiO2 sphere abrasives promote the development of CMP technology and meet the increasing demand for ultra-precision surfaces for industrial applications.” in manuscript and marked in blue. Comments on the Quality of English Language The syntax and grammar is generally ok but proof reading should be conducted. For example in section 2: "Solution A was consisted of..." should be "Solution A was consisting of..."
Re: Thank you for your comment. We have checked the manuscript carefully to avoid grammar, spelling, and other language errors.

Reviewer 2 Report
Comments and Suggestions for Authors
In the proposed paper tilted “Monodisperse SiO2 Spheres Efficient Synthesis and Application in Chemical Mechanical Polishing” titled, J. Ge and co-workers report an improved chemical mechanical polishing (CMP) method carried out by using monodisperse silica (SiOâ‚‚) spheres synthesized via a modified Stöber method with cationic surfactants. These silica particles, are uniform in shape and size (50–150 nm), and have a high surface area (~1155.9 m²/g), enhancing polishing efficiency. When used as abrasives in CMP, particularly CTAB-coated SiOâ‚‚ spheres, they significantly reduced silicon wafer surface roughness, demonstrating their effectiveness in achieving precise, atomic-level polishing.
The manuscript might be interesting and relevant to nanomaterials, nanoengineering, and, more generally, the nanotechnology community; however, it requires numerous changes before publication. Some relevant formal and technical issues must be addressed.
- While the article is generally well written and free of significant grammatical errors, the introduction is brief and lacks depth. The authors are encouraged to revise the manuscript by providing a more comprehensive overview of the research background and current state of the art. Additionally, including more details on the fundamental aspects of the proposed work would enhance the clarity and context of the study.
- The authors should add in the paper’s introduction more information about Silica NPs applications, such as in nanomedicine (https://doi.org/10.1016/j.addr.2023.115115)( https://doi.org/10.1002/wnan.1658), nanotechnology, agriculture (https://doi.org/10.1007/s10311-022-01515-9), coating (https://doi.org/10.1016/j.ceramint.2021.11.239) and so on.
- The introduction should also briefly discuss the characterization techniques used in this study. The authors are advised to mention the role of AFM and SEM in nanoparticle topography analysis, citing relevant reviews or articles (e.g., https://doi.org/10.1016/j.matdes.2024.113036, https://doi.org/10.1002/med.21981). Moreover, the rationale behind choosing these microscopy techniques should be clearly explained.
The introduction does not provide a clear or thorough summary of the key findings. The authors should more explicitly state the novelty and significance of their results. Expanding this section to include a brief discussion of the main outcomes would help readers better appreciate the originality and relevance of the work
- Figure 1 is of low quality and appears distorted. The authors are asked to reproduce this figure at a higher resolution, ensuring correct aspect ratio and clarity.
- In the section “2.4. Characterization” the information about the morphological characterizations with SEM and AFM is very scarce. For example, the acceleration voltage at 15 keV is reported but no other parameters (current, scanning speed, number of pixels, chamber pressure, type of detector used), treatment and preparation of the sample, etc. Even more seriously, all the information about the AFM acquisitions is missing: acquisition method (contact, semicontact, tapping), type of measurements (air, liquid, vacuum), type of tips used, and related parameters such as radius of curvature, elasticity of the cantilever, details on the acquisition parameters (number of pixels, filters...). This lack of information is unacceptable for an article in which AFM and SEM images are reported. The authors must add these details.
- The colors used in the graphs are not very accessible for colourblind readers. Change them using more appropriate palettes (DOI: 10.1002/rth2.12308) in order to improve the accessibility of the manuscript.
- the paper lacks a rigorous statistical analysis of the results. For example, the data reported in Fig. 5 and 6 are completely missing error bars. This suggests that the experiments were not replicated. The authors need to add a statistical analysis of the data.
- The scalebars in the Figure 9 c)-h) are misleading and illegible. The authors should redo them. Also, the z-scale values ​​on the right side of the AFM images should be moved out of the image in order to better observe the surface topography of the sample.
- Why the authors did not characterize the sample surface after the CMP process ?
Author Response
1 While the article is generally well written and free of significant grammatical errors, the introduction is brief and lacks depth. The authors are encouraged to revise the manuscript by providing a more comprehensive overview of the research background and current state of the art. Additionally, including more details on the fundamental aspects of the proposed work would enhance the clarity and context of the study.
Re: Thank you for your suggestion. In Introduction section, we have added the background of SiO2 spheres application, the research background. In the summary, we have stated the significance of this work as following:
The prepared monodisperse SiO2 sphere abrasives promote the development of CMP technology and meet the increasing demand for ultra-precision surfaces for industrial applications.
2 The authors should add in the paper’s introduction more information about Silica NPs applications, such as in nanomedicine (https://doi.org/10.1016/j.addr.2023.115115)( https://doi.org/10.1002/wnan.1658), nanotechnology, agriculture (https://doi.org/10.1007/s10311-022-01515-9), coating (https://doi.org/10.1016/j.ceramint.2021.11.239) and so on.
Re: Thank you for your comments. SiO2 spheres have been applied in many fields, such as nanomedicine, coating, excipients and additives[12-14]. In addition, the narrow diameter distribution of monodisperse SiO2 spheres can reduce the agglomeration of abrasives so that the SiO2 sphere can be used as abrasives to reduce the surface roughness of fine instruments[15].
3 The introduction should also briefly discuss the characterization techniques used in this study. The authors are advised to mention the role of AFM and SEM in nanoparticle topography analysis, citing relevant reviews or articles (e.g., https://doi.org/10.1016/j.matdes.2024.113036, https://doi.org/10.1002/med.21981). Moreover, the rationale behind choosing these microscopy techniques should be clearly explained.
Re: Thank you for your suggestion. We have added the brief discussion of SEM and AFM characterization techniques in the introduction section as following:
The surface topography can be detected via SEM and AFM characterization techniques. The physical properties and surface topography of spheres can be obtained by SEM[20]. For AFM, the surface morphologies provided by the interaction force between the probe tip and surface[21]. The synthesis, development and characterization of nanoparticle in the field of surface and interface.
4 The introduction does not provide a clear or thorough summary of the key findings. The authors should more explicitly state the novelty and significance of their results. Expanding this section to include a brief discussion of the main outcomes would help readers better appreciate the originality and relevance of the work
Re: Thank you for your comment. We have added related description in introduction section to state the novelty and significance of this work as following:
The synthesis, development and characterization of nanoparticle in the field of surface and interface.
The obtained SiO2 spheres used as a abrasives is conducive to the surface modifying and fine processing.
5 Figure 1 is of low quality and appears distorted. The authors are asked to reproduce this figure at a higher resolution, ensuring correct aspect ratio and clarity.
Re: Thank you for your suggestion. We have reproduced Figure 1 at a higher resolution.
6 In the section “2.4. Characterization” the information about the morphological characterizations with SEM and AFM is very scarce. For example, the acceleration voltage at 15 keV is reported but no other parameters (current, scanning speed, number of pixels, chamber pressure, type of detector used), treatment and preparation of the sample, etc. Even more seriously, all the information about the AFM acquisitions is missing: acquisition method (contact, semicontact, tapping), type of measurements (air, liquid, vacuum), type of tips used, and related parameters such as radius of curvature, elasticity of the cantilever, details on the acquisition parameters (number of pixels, filters...). This lack of information is unacceptable for an article in which AFM and SEM images are reported. The authors must add these details.
Re: Thank you for your comment. We have added related description in manuscript as following: The structure and morphology of the samples were investigated with a Zeiss Sigma 300 scanning electron microscope at an acceleration voltage of 15 kV and the probe current was 50 pA. The morphology and surface roughness of the polished silicon wafers were characterized using atomic force microscopy with Multimode 8, Bruker, Santa Barbara, CA. The samples were measured by contact mode in air. The tip radius was 10 nm and the tip height was 17.5 μm.
7 The colors used in the graphs are not very accessible for colourblind readers. Change them using more appropriate palettes (DOI: 10.1002/rth2.12308) in order to improve the accessibility of the manuscript.
Re: Thank you for your comment. We have changed the colors in the graphs using more appropriate palettes to improve the accessibility of the manuscript.
8 the paper lacks a rigorous statistical analysis of the results. For example, the data reported in Fig. 5 and 6 are completely missing error bars. This suggests that the experiments were not replicated. The authors need to add a statistical analysis of the data.
Re: The BET data were presented in Figure 5 and 6. Generally, it is not necessary to conduct multiple experiments to take the average value for BET data. Similar processing methods also exist in the high-level papers that have been published.
[1] Jiang J. Wang H. Lin J. Wang F. Liu Z, Wang L, Li Z, Li Y, Li Y. Nature-inspired hierarchical building materials with low CO2 emission and superior performance. Nature communications 2025, 16, 3018.
[2] Yang C, Luo L, Zhao T, Cao J, Lin Q. Chitosan-based porous carbon material with lamellar stacking structure for efficient CO2 adsorption and catalytic conversion. Chemical Engineering Journal 2025,512, 162597.
[3] Xia Z, Zhang X, Shen L, Wang N, Wang T, Wu L. Design of freestanding flexible electrodes with hierarchical porous structure from cotton textiles for high-performance supercapacitors. Journal of Energy Storage 2025, 116, 115972.
[4] Yan W, Sun Z, Dong K, Guo F. Effects of dry and wet torrefaction pretreatment on the physicochemical structure of corn stover-derived porous carbon and its performance in supercapacitor applications. Biomass and Bioenergy 2025, 197, 107773.
[5] Duan L, Fan J, Chen G, Qiu P, Zhang X, Zhai X. Synthesis of porous PEGDMA microspheres via suspension polymerization and the effect of different porogens on the porous structure. Colloids and Surfaces A: Physicochemical and Engineering Aspects 2025, 709, 136125.
9 The scalebars in the Figure 9 c)-h) are misleading and illegible. The authors should redo them. Also, the z-scale values ​​on the right side of the AFM images should be moved out of the image in order to better observe the surface topography of the sample.
Re: Thank you for your suggestion. We have redone the Figure 9 and 10 to correct these mistakes.
10 Why the authors did not characterize the sample surface after the CMP process?
Re: Thank you for your comment. The sample surface morphology after the CMP process was evaluated via AFM. In this work, the main research purpose is to achieve the application of SiO2 nanospheres in the CMP process for surface fine processing.

Round 2
Reviewer 2 Report
Comments and Suggestions for Authors
I appreciate the revisions made to the manuscript that have increased the clarity of the paper.
In my opinion, the paper coul be accept for the publication in the present form.